# Exploring bat-inspired cyclic tryptophan diketopiperazines as ABCB1 Inhibitors
Javier Yu Peng Koh [1,8], Yoko Itahana [1,8], Alexander Krah [2,8], Habib Mostafa[3,8], Mingmin Ong[1], Sahana Iwamura[4], Dona Mariya Vincent [3], Sabhashina Radha Krishnan [1], Weiying Ye [1], Pierre Wing Chi Yim[1], Tushar M. Khopade[3], Kunihiko Chen[1], Pui San Kong[5], Lin-Fa Wang [5], Roderick W. Bates [6], Yasuhisa Kimura [4], Rajesh Viswanathan[3,9] ✉, Peter J. Bond [2,7,9] ✉ & Koji Itahana [1,9] ✉

Chemotherapy-induced drug resistance remains a major cause of cancer recurrence and patient mortality. ATP binding cassette subfamily B member 1 (ABCB1) transporter overexpression in tumors contributes to resistance, yet current ABCB1 inhibitors have been unsuccessful in clinical trials. To address this challenge, we propose a new strategy using tryptophan as a lead molecule for developing ABCB1 inhibitors. Our idea stems from our studies on bat cells, as bats have low cancer incidences and high ABCB1 expression. We hypothesized that potential ABCB1 substrates in bats could act as competitive inhibitors in humans. By molecular simulations of ABCB1-substrate interactions, we generated a benzylated *Cyclo*-tryptophan (C3N-Dbn-Trp2) that inhibits ABCB1 activity with efficacy comparable to or better than the classical inhibitor, verapamil. C3N-Dbn-Trp2 restored chemotherapy sensitivity in drug-resistant human cancer cells with no adverse effect on cell proliferation. Our unique approach presents a promising lead toward developing effective ABCB1 inhibitors to treat drug-resistant cancers.

The development of drug resistance in cancer cells is a primary reason for cancer relapse and eventual cancer patient death. This underscores the critical need for new strategies to overcome drug resistance in cancer treatment. Most cancer research utilizes cancer-prone, short-lived models such as mice and rats. However, long-lived mammals have emerged to possess unique anticancer strategies that could potentially offer new treatment options[1]. Bats are small, long-lived mammals with an extremely low incidence of cancer[1,2]. We recently found that genotoxic drugs caused significantly less DNA damage in bat cells than in human and mouse cells. We reported that ubiquitous and high expression of the ATP-binding cassette (ABC) transporter ABCB1/MDR1(multi-drug resistance protein 1)/P-glycoprotein in bats leads to drug efflux, minimizing drug accumulation and DNA damage, indicating the possible involvement of ABCB1 in tumour suppression in bats[3].

In humans, ABCB1 expression is limited to tissues for detoxification such as the kidney, intestine, liver, hematopoietic stem cells, and blood-brain barrier to protect important organs and cells from harmful compounds[3–5]. Consistent with this, a double knockout of *Abcb1a* and *Abcb1b* in mice shows extremely high sensitivity to toxic chemicals[6]. However, cancers often take advantage of these detoxifying properties of ABCB1 and upregulate its expression in response to chemotherapy. Although drug resistance in cancer involves multiple mechanisms, including mutations and altered gene expression, ABCB1 overexpression is one of the important mechanisms of drug resistance and cancer relapse[4,7]. Many publications on ABCB1-mediated drug resistance and previous efforts in developing ABCB1 inhibitors highlight the importance of ABCB1-mediated drug efflux in drug resistance in cancers[8–13]. ABCB1 overexpression in cancers has often been associated with poor patient prognosis. In addition, overexpression of ABCB1 is not

[1]Programme in Cancer & Stem Cell Biology, Duke-NUS Medical School, Singapore, Singapore. [2]Bioinformatics Institute (BII), Agency for Science, Technology, and Research (A*STAR), Singapore, Singapore. [3]Department of Chemistry, Indian Institute of Science Education and Research (IISER) Tirupati, Andhra Pradesh, India. [4]Division of Applied Life Sciences, Graduate School of Agriculture, Kyoto University, Kyoto, Japan. [5]Programme in Emerging Infectious Diseases, Duke-NUS Medical School, Singapore, Singapore. [6]School of Chemistry, Chemical Engineering and Biotechnology, Nanyang Technological University, Singapore, Singapore. [7]Department of Biological Sciences, National University of Singapore, Singapore, Singapore. [8]These authors contributed equally: Javier Yu Peng Koh, Yoko Itahana, Alexander Krah, Habib Mostafa.[9]These authors jointly supervised this work: Rajesh Viswanathan, Peter J. Bond, Koji Itahana. ✉e-mail: rajesh@iisertirupati.ac.in; peterjb@bii.a-star.edu.sg; koji.itahana@duke-nus.edu.sg

limited to specific cancer types, and ABCB1 inhibitors could therefore contribute to the treatment of a broad range of cancers. First-generation ABCB1 inhibitors such as verapamil and cyclosporine A, which were developed primarily for other indications[14], have been observed to act as substrates and competitive inhibitors of ABCB1[15]. However, these compounds require high doses to inhibit ABCB1 and have failed in clinical trials due to toxicity[14,15]. Improvements to these first-generation inhibitors led to second-generation inhibitors such as dexverapamil and PSC 833. These chemicals improved affinity for ABCB1, but failed in clinical trials due to toxicity and unpredictable pharmacokinetic responses[16]. The third-generation inhibitors, tariquidar, VX-710 and GF120918, were found by high-throughput screening using chemically modified ABCB1 substrates. Although they achieved higher ABCB1 inhibitory effects, they failed in clinical trials due to multiple reasons including non-negligible off-target effects[17]. Given that mice with a double knockout of *Abcb1a* and *Abcb1b* showed no physiological abnormalities in the absence of toxins[6], ABCB1 inhibition itself may have minimal adverse effects. This suggests that ABCB1 inhibitors, if highly selective, could potentially be well-tolerated. The failures of synthetic compounds in clinical trials have prompted research into natural ABCB1 inhibitors, which might be safer and less toxic.

In 2018, the first structure of a human-mouse ABCB1 chimaera bound to zosuquidar, an inhibitor, was resolved by cryogenic electron microscopy (cryo-EM), showing two zosuquidar molecules bound to ABCB1[18]. Subsequently, the ATP-bound outward-facing conformation of human ABCB1 was first reported using cryo-EM[19]. In 2019, the first structural data for human ABCB1 interacting with a substrate, paclitaxel, and with an inhibitor, zosuquidar, were reported using cryo-EM or double electron-electron resonance (DEER)[20,21]. In 2020, another cryo-EM analysis of the ABCB1 structure identified distinct binding sites for substrates and inhibitors to ABCB1[22]. This study revealed that ABCB1 substrates bind only to the central cavity, while ABCB1 substrates that can act as competitive inhibitors occupy both the central cavity and the access tunnel in the ABCB1 structure, which is critical for ABCB1 inhibition. These recent structural reports help researchers understand the mechanism by which ABCB1 inhibitors inhibit ABCB1-mediated efflux and may enable the design of ABCB1 inhibitors assisted by computational modelling to optimize the binding between ABCB1 and inhibitors.

Interestingly, a recent report showed that ABCB1 undergoes positive selection among 18 bat species[23], suggesting the importance of ABCB1 in the evolution of bats which are the only mammals that can fly. As ABCB1 is ubiquitously and highly expressed in many tissues in bats[3], besides the protection from xenobiotics, high ABCB1 expression might be used for the efflux of metabolic by-products due to their flight, in order to maintain cellular homeostasis. These unique metabolites effluxed by ABCB1 in bats may be used to design safer competitive inhibitors of ABCB1 against drug-resistant human cancers, as they are derived from natural metabolites. Competitive inhibition is one of the common ways to inhibit transporter activity in cells. Competitive inhibition occurs when a molecule binds to the transporter's substrate binding site and prevents the substrate from binding.

Using mass spectrometry-based metabolic analysis, we discovered that knockout of *ABCB1* in bat cells leads to tryptophan accumulation in cells. Furthermore, we showed that tryptophan derivatives, but not the derivatives of other amino acids, have ABCB1 inhibitory activity. Several tryptophan derivatives designed by structure-guided computational modelling of human ABCB1-ligand binding efficiently inhibited ABCB1 in human cancer cells and sensitised them to chemotherapeutic agents. Based on these studies, we propose a unique strategy in which tryptophan, a natural metabolite, could be a promising lead molecule for the development of ABCB1 inhibitors with minimal adverse effects. In this report, we demonstrate the feasibility of this unique strategy by designing a derivative of dimeric tryptophan diketopiperazine that displays impressive ABCB1 inhibition against human cancer cells with no obvious toxicity to cells in culture.

## Results

### Tryptophan accumulates in *ABCB1*-knockout PaKiT03 cells

We reported that, unlike humans, bats express high levels of ABCB1 ubiquitously in their tissues and that high ABCB1 expression protects bat cells from DNA damage caused by genotoxic chemicals[3]. We reasoned that the ubiquitously elevated ABCB1 expression in bats also reflects the routine efflux of cellular substrates to maintain homeostasis. Therefore, we initially explored the identification of endogenous substrates of ABCB1. We first established four *ABCB1* CRISPR-knockout (KO) clones of *Pteropus alecto* (*P. alecto*)-derived PaKiT03 kidney cells. These clones were verified by Western blotting analysis for their loss of protein expression (Fig. 1a). We incubated cells with doxorubicin, a chemotherapeutic agent and a well-known fluorescent substrate of ABCB1 that accumulates in cells upon ABCB1 inhibition. As expected, accumulation of doxorubicin was observed in PaKiT03 parental cells when treated with verapamil (a known ABCB1 inhibitor) (Fig. 1b). PaKiT03 *ABCB1* KO cell clones showed accumulation of doxorubicin regardless of the treatment with verapamil, to levels similar to those in PaKiT03 parental cells treated with verapamil (Fig. 1b), confirming the loss of efflux activity of ABCB1 in PaKiT03 CRISPR KO clones. We then conducted metabolic profiling of these cells using a mass-spectrometry-based approach. We speculated that potential ABCB1 substrates could accumulate more in *ABCB1* KO cells. Ingenuity Pathway Analysis demonstrated that downregulation of the tryptophan degradation pathway was the most consistent feature in all four *ABCB1* KO PaKiT03 cells (Fig. 1c), suggesting the possible accumulation of tryptophan in *ABCB1* KO cells. In fact, *ABCB1* KO cells displayed elevated intracellular tryptophan compared to parental cells (Fig. 1d). These results suggest that tryptophan is a potential cellular substrate candidate for ABCB1.

### Tryptophan structure is essential for ABCB1 inhibition

ABC transporter-mediated transport is driven by ATP hydrolysis. Therefore, we performed in vitro ATPase assay to test whether tryptophan is transported through ABCB1. We used human recombinant ABCB1 protein because protein sequences of ABCB1 are highly conserved between bats and humans[3]. Even at 25 mM, which is much higher than the intracellular concentration of tryptophan (approximately 60 μM[24]), no stimulation of ATPase activity by tryptophan was observed using human ABCB1 (Fig. 2a), suggesting that tryptophan itself is unlikely to be a substrate for human ABCB1. As tryptophan is a small molecule compared to most of the known ABCB1 substrates[25], we next examined whether *Cyclo*-(L-Trp-L-Trp) can be recognized by ABCB1. *Cyclo*-(L-Trp-L-Trp) is a cyclic dimer of tryptophan and is naturally produced by bacteria[26–28] (Supplementary Fig. S1). We detected stimulation of ATPase activity by *Cyclo*-(L-Trp-L-Trp) at 2 mM, indicating that *Cyclo*-(L-Trp-L-Trp) is recognized by ABCB1. These results suggest that *Cyclo*-(L-Trp-L-Trp) could be a potential ABCB1 substrate, although the activity was still lower than using a known ABCB1 substrate, verapamil (Fig. 2a).

As the majority of known ABCB1 inhibitors are also ABCB1 substrates and act as competitive inhibitors[11], we speculated that *Cyclo*-(L-Trp-L-Trp) can be a competitive inhibitor of ABCB1. Verapamil is one such example. To test this hypothesis, we pre-treated bat PaKiT03 cells with an increasing dose of L-tryptophan or *Cyclo*-(L-Trp-L-Trp) for 30 minutes and then co-incubated them with rhodamine 123 (Rh123), a well-known fluorescent substrate of ABCB1, for 3 hours. We examined by flow cytometry whether these tryptophan-derivative compounds prevent the efflux of rhodamine 123. We observed that rhodamine 123 was gradually accumulated in PaKiT03 cells by increasing the dose of tryptophan up to 20 mM (Fig. 2b), suggesting that tryptophan may act as a competitive inhibitor of ABCB1 even though it is not transported by ABCB1. On the other hand, lower concentrations of *Cyclo*-(L-Trp-L-Trp) (1 mM) compared to tryptophan (20 mM) were sufficient to trigger rhodamine 123 accumulation in PaKiT03 cells (Fig. 2c). Rhodamine 123 was accumulated by 1 mM of *Cyclo*-(L-Trp-L-Trp) up to half of the amounts of those in PaKiT03 cells treated with 5 μM of verapamil (Fig. 2c). Similar cyclic dipeptide compounds containing only one or no tryptophan, *Cyclo*-(L-Leu-L-Trp), *Cyclo*-(L-Trp-L-Pro), and

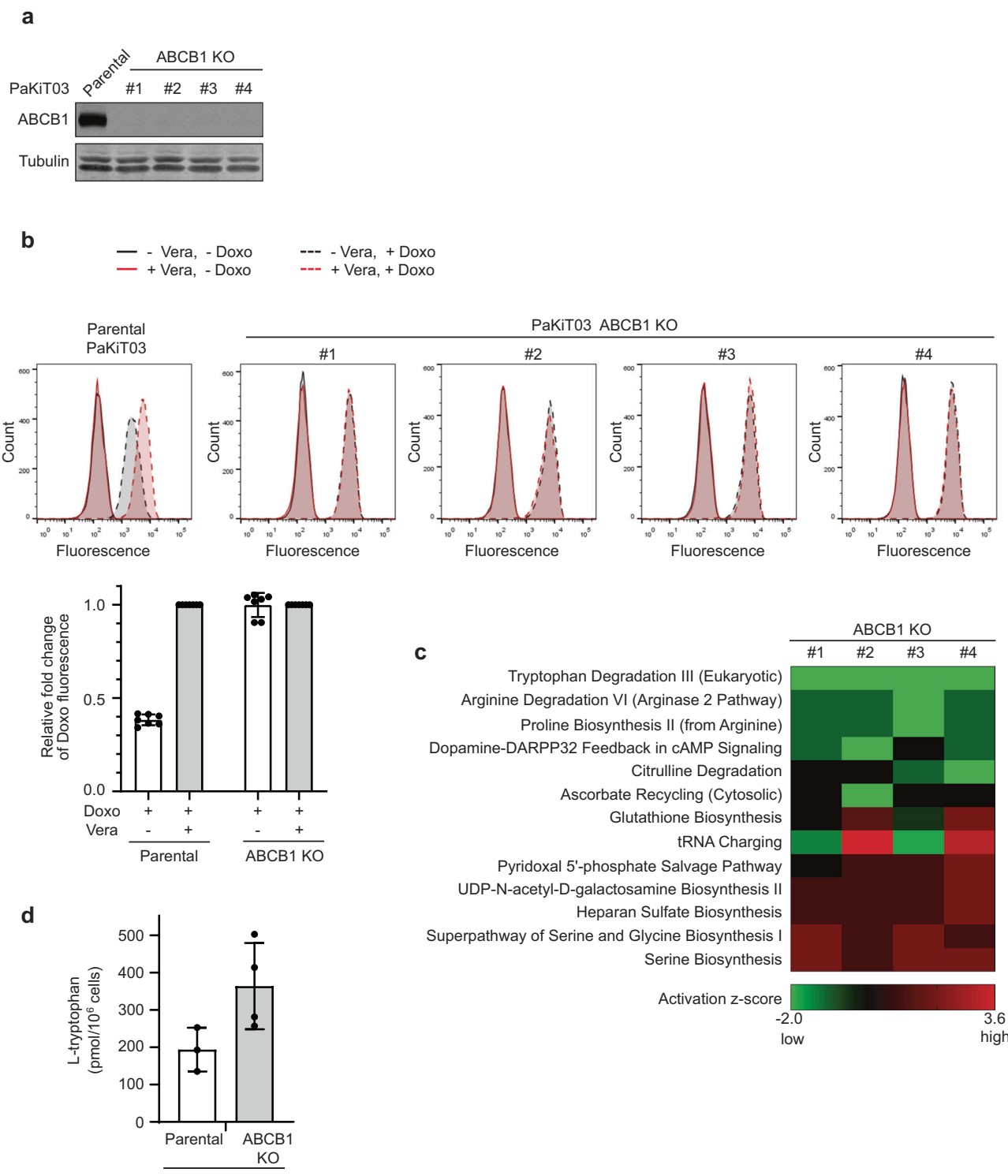

**Fig. 1 | Metabolomics analysis of *P. alecto*-derived PaKiT03 cells. a** Western blot analysis of ABCB1 in parental PaKiT03 cells and *ABCB1* knockout (KO) clones. Tubulin was used as a loading control. **b** Flow cytometry analysis of doxorubicin (Doxo) accumulation in parental PaKiT03 cells and *ABCB1* KO clones. Cells were incubated with or without 10 μM Doxo. 5 μM verapamil (Vera) was used to inhibit ABCB1 efflux activity maximally. The bar graph represents the mean ( ± SD of seven independent experiments) fluorescent intensity of Doxo relative to those treated with both Doxo and Vera in each cell line (the values for lanes 2 and 4 were set to 1, respectively). Black solid line: without Vera without Doxo; Black dashed line: without Vera with Doxo; Red solid line: with Vera without Doxo; Red dashed line: with Vera with Doxo. **c** Heat map of metabolic pathways in PaKiT03 *ABCB1* KO clones compared to parental cells using Ingenuity Pathway Analysis platform. **d** Intracellular L-tryptophan amounts in parental PaKiT03 (n = 3) and *ABCB1* KO (4 clones, *n* = 1) cells. The bar graph represents the mean ± SD.

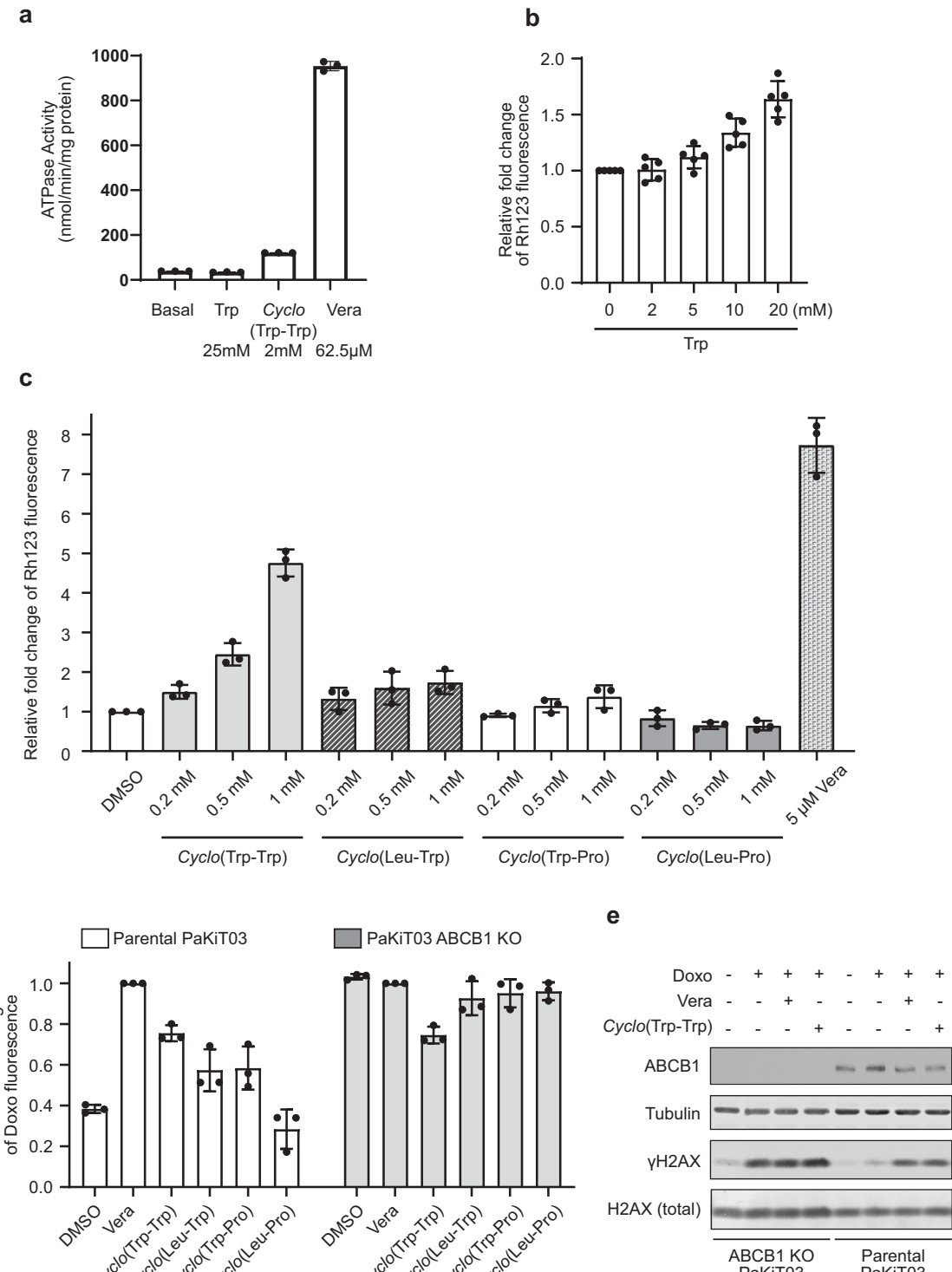

**Fig. 2 | ABCB1 inhibition by *Cyclo*-(L-Trp-L-Trp) in PaKiT03 cells. a** ATPase assay of purified and reconstituted human ABCB1 protein with tryptophan (Trp), *Cyclo*-(L-Trp-L-Trp), or verapamil (Vera) at the indicated concentration. Bar graphs represent the mean ( ± SD of three independent experiments). **b** Flow cytometry analysis of rhodamine 123 (Rh123) accumulation. Parental PaKiT03 cells were pre-treated with the indicated dose of Trp before incubating with 2.5 μM Rh123. Bar graphs represent the mean ( ± SD of five independent experiments) fluorescent intensity of Rh123 relative to the control (DMSO, the first bar). **c** Flow cytometry analysis of rhodamine 123 (Rh123) accumulation. Parental PaKiT03 cells were pre-treated with the indicated concentration of *Cyclo*-(L-Trp-L-Trp), *Cyclo*-(L-Lue-L-Trp), *Cyclo*-(L-Trp-L-Pro), *Cyclo*-(L-Leu-L-Pro), or verapamil (Vera) before incubating with 2.5 μM Rh123. Bar graphs represent the mean ( ± SD of three

independent experiments) fluorescent intensity of Rh123 relative to the control (DMSO, the first bar). **d** Flow cytometry analysis of doxorubicin (Doxo) accumulation. Parental PaKiT03 cells and *ABCB1* KO clones were pre-treated with or without 5 μM Vera or 1 mM of *Cyclo*-(L-Trp-L-Trp), *Cyclo*-(L-Leu-L-Trp), *Cyclo*-(L-Trp-L-Pro), or *Cyclo*-(L-Leu-L-Pro) before incubating with 10 μM Doxo. Bar graphs represent the mean ( ± SD of three independent experiments) fluorescent intensity of Doxo relative to the Vera-treated cells (Vera, the second bar in each cell line). White bars: parental PaKiT03 cells; Gray bars: PaKiT03 *ABCB1* KO cells. **e** Western blot analysis of DNA damage response. Parental PaKiT03 and *ABCB1* KO cells were pre-treated with or without 5 μM Vera or 1 mM *Cyclo*-(L-Trp-L-Trp) before incubating with 0.3 μM Doxo for an additional 3 hours. Tubulin was used as a loading control.

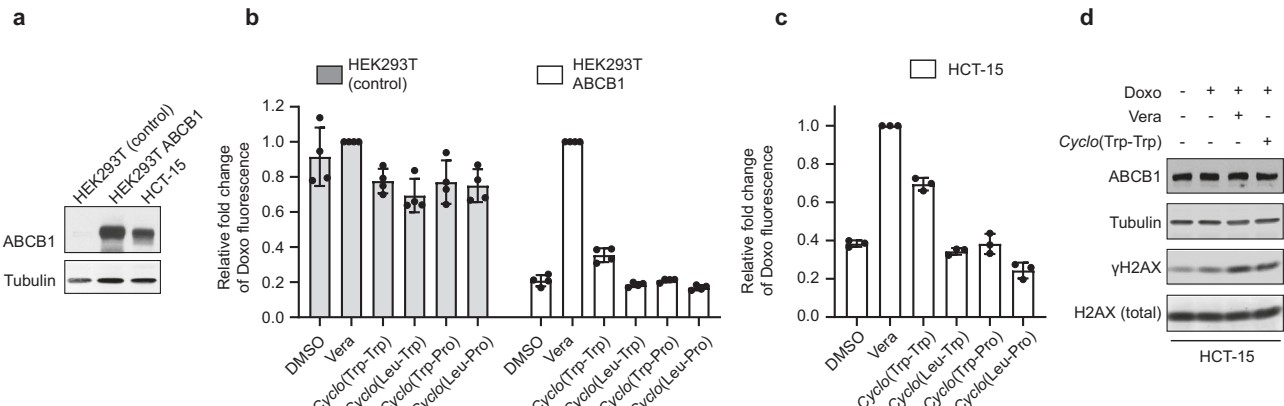

**Fig. 3 | ABCB1 inhibition by *Cyclo*-(L-Trp-L-Trp) in human cells. a** Western blot analysis of ABCB1 expression. Proteins were extracted from HEK293T (control), HEK293T ABCB1 (the cells stably expressing exogenous human ABCB1), and HCT-15 cells. Tubulin was used as a loading control. **b** Flow cytometry analysis of doxorubicin (Doxo) accumulation. HEK293T control or ABCB1 expressing cells were pre-treated with or without 5 μM verapamil (Vera) or 1 mM of *Cyclo*-(L-Trp-L-Trp), *Cyclo*-(L-Leu-L-Trp), *Cyclo*-(L-Trp-L-Pro), or *Cyclo*-(L-Leu-L-Pro) before incubating with 10 μM Doxo. Bar graphs represent the mean ( ± SD of four independent experiments) fluorescent intensity of Doxo relative to the Vera-treated cells (Vera, the second bar in each cell line). Gray bars: HEK293T cells; white bars:

HEK293T ABCB1 expressing cells. **c** Flow cytometry analysis of doxorubicin (Doxo) accumulation. HCT-15 cells were pre-treated with or without 5 μM Vera or 1 mM of *Cyclo*-(L-Trp-L-Trp), *Cyclo*-(L-Leu-L-Trp), *Cyclo*-(L-Trp-L-Pro), or *Cyclo*-(L-Leu-L-Pro) before incubating with 10 μM Doxo. Bar graphs represent the mean ( ± SD of three independent experiments) fluorescent intensity of Doxo relative to the Vera-treated cells (Vera, the second bar). **d** Western blot analysis of DNA damage response. HCT-15 cells were pre-treated with or without 5 μM Vera or 1 mM *Cyclo*-(L-Trp-L-Trp) before incubating with 0.3 μM Doxo for an additional 3 hours. Tubulin was used as a loading control.

*Cyclo*-(L-Leu-L-Pro), resulted in less or no rhodamine 123 accumulation at 1 mM (Fig. 2c and Supplementary Fig. S1), suggesting that the structure of tryptophan is critical for inhibiting the efflux of rhodamine 123. Consistent with these results, another ABCB1 substrate, doxorubicin, was also accumulated by *Cyclo*-(L-Trp-L-Trp) (Fig. 2d). Doxorubicin accumulation was less affected by *Cyclo*-(L-Leu-L-Trp) or *Cyclo*-(L-Trp-L-Pro) treatment and not affected by *Cyclo*-(L-Leu-L-Pro) that does not contain a tryptophan structure. *ABCB1* KO PaKiT03 cells consistently accumulated high levels of doxorubicin regardless of the treatments (Fig. 2d), suggesting that increased doxorubicin accumulation by *Cyclo*-(L-Trp-L-Trp) treatment in parental PaKiT03 cells is due to the inhibition of ABCB1 by *Cyclo*-(L-Trp-L-Trp). Doxorubicin is a chemotherapeutic agent used to induce DNA damage, which can be monitored with a DNA double-strand break marker, phosphorylated histone 2 A variant X (γH2AX). To monitor the effect of *Cyclo*-(L-Trp-L-Trp)-mediated inhibition of ABCB1 on DNA damage induced by doxorubicin, we pretreated PaKiT03 cells with DMSO (control), verapamil, or *Cyclo*-(L-Trp-L-Trp) for 30 minutes, and then added doxorubicin for 3 hours. Doxorubicin treatment alone slightly induced γH2AX in parental PaKiT03 cells (Fig. 2e). Pretreatment with verapamil or *Cyclo*-(L-Trp-L-Trp) further increased γH2AX by doxorubicin in parental PaKiT03 cells (Fig. 2e), suggesting that both verapamil and *Cyclo*-(L-Trp-L-Trp) increased DNA damage by inhibiting ABCB1-mediated doxorubicin efflux. γH2AX was constantly high in *ABCB1* KO PaKiT03 cells with or without pretreatment of verapamil or *Cyclo*-(L-Trp-L-Trp) (Fig. 2e), consistent with the high accumulation of doxorubicin regardless of the treatment in these cells (Fig. 2d). Together, these results suggest that the structure of tryptophan, or its cyclic diketopiperazine structure, is key to the inhibition of bat ABCB1 in PaKiT03 cells.

### *Cyclo*-(L-Trp-L-Trp) inhibits human ABCB1
As stated above, bat and human ABCB1 protein sequences are approximately 90% identical[3]. We next determined if *Cyclo*-(L-Trp-L-Trp) inhibits human ABCB1 in human cell lines. We generated human HEK293T (embryonic kidney cell transformed with SV40 large T antigen) cells stably expressing human ABCB1 (Fig. 3a). Parental HEK293T cells do not express ABCB1, and therefore, doxorubicin was accumulated in the cells regardless of the treatments with verapamil or cyclo-dipeptide compounds (Fig. 3b). On the other hand, ABCB1-expressing HEK293T cells effluxed doxorubicin

unless the cells were pre-treated with verapamil (Fig. 3b). Although the effect was not as obvious as verapamil, *Cyclo*-(L-Trp-L-Trp) also prevented the efflux of doxorubicin compared to the non-treated (DMSO) cells in ABCB1-expressing HEK293T cells (Fig. 3b). *Cyclo*-(L-Leu-L-Trp), *Cyclo*-(L-Trp-L-Pro), and *Cyclo*-(L-Leu-L-Pro) did not have any effects on doxorubicin amounts in ABCB1-expressing HEK293T cells (Fig. 3b), confirming that the structure of two tryptophan is important to inhibit ABCB1. We also used human HCT-15 (colorectal adenocarcinoma) cells that endogenously express ABCB1 (Fig. 3a). Consistent with the results with HEK293T cells, *Cyclo*-(L-Trp-L-Trp), but not *Cyclo*-(L-Leu-L-Trp), *Cyclo*-(L-Trp-L-Pro), or *Cyclo*-(L-Leu-L-Pro), increased doxorubicin accumulation compared to the non-treated (DMSO) cells (Fig. 3c). *Cyclo*-(L-Trp-L-Trp) treatment also increased doxorubicin-induced γH2AX similar to verapamil treatment (Fig. 3d). Thus, these results indicate that *Cyclo*-(L-Trp-L-Trp) inhibits not only bat ABCB1 but also human ABCB1.

### Computational simulation identifies the improved stability of *Cyclo*-(L-1-methyl-Trp-L-1-methyl-Trp) in the pocket of ABCB1
We next investigated the possible binding sites of *Cyclo*-(L-Trp-L-Trp) within human ABCB1 using computational molecular modelling. We thus conducted molecular docking followed by simulations of the binding mode for a series of compounds with ABCB1 embedded within a realistic membrane model (Supplementary Fig. S2a). It should be noted that for such large, bulky hydrophobic ligands being investigated here, multiple binding poses, rather than just a single specific state, may be responsible for their inhibitory activity, particularly given the complexity of ABCB1 and its promiscuous substrate binding, and thus our computational approach served as an exploratory guide for possible modes of interaction. According to published cryo-EM structures, paclitaxel, a substrate of ABCB1, binds to the transporter as a single molecule (PDB-ID: 6QEX[21]). In contrast, zosuquidar, a substrate with competitive inhibition activity, binds to ABCB1 as two molecules (PDB-ID: 7A6f[22]) (Supplementary Fig. S2b). Initially, as a negative control, computational modelling was performed for *Cyclo*-(L-Leu-L-Pro), indicating that two molecules do not bind stably to the central pocket (Supplementary Fig. S2b, Table 1), consistent with the experimentally demonstrated inability of *Cyclo*-(L-Leu-L-Pro) to inhibit ABCB1 (Fig. 3b, c). Furthermore, our modelling suggested that two *Cyclo*-(L-Trp-L-Trp) molecules may bind to ABCB1 (Fig. 4a, b). Similar to the reported

**Table 1 | Molecular interactions of the compounds during molecular dynamics simulations**

| System | Number of hydrogen bonds (protein-ligand) | Number of hydrogen bonds (ligand-ligand) | Number of hydrophobic contacts (protein-ligand) | Number of hydrophobic contacts (ligand-ligand) | Total binding free energy (kcal/mol) |
|---|---|---|---|---|---|
| Cyclo-(L-Leu-L-Pro) | 1.3 ± 0.9 | 0.1 ± 0.3 | 22.3 ± 4.7 | 0.8 ± 1.6 | −33.7 ± 2.0 |
| Cyclo-(L-Trp-L-Trp) | 3.2 ± 1.3 | 0.3 ± 0.5 | 38.0 ± 6.1 | 2.2 ± 2.3 | −46.9 ± 3.1 |
| Cyclo-(L-1-methyl-Trp-L-1-methyl-Trp) | 2.5 ± 1.1 | 0 ± 0 | 40.5 ± 5.6 | 0.6 ± 1.0 | −56.2 ± 1.7 |
| C3N-Dbn-Trp2 | 2.4 ± 0.9 | 0 ± 0 | 46.7 ± 6.8 | 6.8 ± 3.2 | −62.7 ± 0.7 |

The mean number of hydrogen bonds and hydrophobic contacts between ligand and protein and between both ligands are shown, along with the binding energies based on the Molecular Mechanics Poisson-Boltzmann Surface Area (MM-PBSA) calculations.

substrates with competitive inhibition activity[22], one molecule (Fig. 4b, ligand 1 in orange) was predicted to occupy the central pocket, with another (Fig. 4b, ligand 2 in magenta) bound to the access tunnel which may be critical for ABCB1 inhibition[22]. It was demonstrated that molecules that only occupy the central pocket of ABCB1 act as substrates whereas substrates with inhibitory activities occupy not only the central pocket but also the access tunnel[22]. Several ABCB1 inhibitors were reported to be surrounded by hydrophobic residues, in particular phenylalanines, including F239, F303, and F994 in the access tunnel[22]. Based on our calculations, ligand 2 of *Cyclo-(L-Trp-L-Trp)* was also surrounded by F239, F303, and F994 (Fig. 4c), suggesting it is a promising lead molecule for developing ABCB1 inhibitors. Because ABCB1 substrates tend to be hydrophobic molecules[25], we added methyl groups at the Nε1 position of both tryptophan side chains to increase the hydrophobicity and performed computational modelling of methylated *Cyclo-(L-Trp-L-Trp)*, *Cyclo-(L-1-methyl-Trp-L-1-methyl-Trp)* (Fig. 4d, e). This addition of methyl groups was hypothesized to increase the stability of ligand 2 in the hydrophobic, phenylalanine-rich access tunnel. Our modelling confirmed that, similar to *Cyclo-(L-Trp-L-Trp)*, *Cyclo-(L-1-methyl-Trp-L-1-methyl-Trp)* binds to both the central pocket and the access tunnel (Fig. 4e, ligand 1 in orange and ligand 2 in magenta). We noticed that the new 1-methyl group is now very close to hydrophobic F994 residue (Fig. 4f). Furthermore, on average across multiple simulation replicas, *Cyclo-(L-1-methyl-Trp-L-1-methyl-Trp)* fitted ABCB1 more favorably than *Cyclo-(L-Trp-L-Trp)* (Table 1). To quantitatively assess the simulated binding poses, we carried out Molecular Mechanics Poisson-Boltzmann Surface Area (MM-PBSA) calculations. These data indicate that cyclo-(L-Leu-L-Pro) binds with lowest affinity to ABCB1, followed by cyclo-(L-Trp-L-Trp), and finally *Cyclo-(L-1-methyl-Trp-L-1-methyl-Trp)* has the strongest binding affinity (Table 1 and S1). Consistent with these results, we also observed that Cyclo-(L-1-methyl-Trp-L-1-methyl-Trp) consumed larger amounts of ATP than *Cyclo-(L-Trp-L-Trp)* in the in vitro ATPase assay (Fig. 4g), suggesting that human ABCB1 interacts with *Cyclo-(L-1-methyl-Trp-L-1-methyl-Trp)* more efficiently than *Cyclo-(L-Trp-L-Trp)*.

**1-methyl modification enhances the efficacy of ABCB1 inhibition and doxorubicin-induced cell death**

To further verify our computational predictions and in vitro ATPase assay results, we tested *Cyclo-(L-1-methyl-Trp-L-1-methyl-Trp)* for ABCB1 inhibition by flow cytometry. PaKiT03 parental and *ABCB1* KO cells were pretreated with an increasing dose of *Cyclo-(L-1-methyl-Trp-L-1-methyl-Trp)* for 30 minutes and then co-incubated with doxorubicin for 3 hours. *Cyclo-(L-1-methyl-Trp-L-1-methyl-Trp)* started inhibiting ABCB1 efflux activity at a lower concentration (0.03 mM) compared to *Cyclo-(L-Trp-L-Trp)* (1 mM) in PaKiT03 cells (Fig. 5a). We also tested *Cyclo-(L-1-methyl-Trp-L-1-methyl-Trp)* for human HEK293T ABCB1-expressing cells and observed dose-dependent ABCB1 inhibition (Fig. 5b). On the other hand, cells that do not express ABCB1 accumulated doxorubicin regardless of the treatments (Fig. 5a, b). Similarly, human HCT-15 cells (endogenous ABCB1 expressing cells) exhibited doxorubicin accumulation by *Cyclo-(L-1-methyl-Trp-L-1-methyl-Trp)* in a dose-dependent manner (Fig. 5c). These results suggested that consistent with our simulations, 1-methyl modification enhances the efficacy of ABCB1 inhibition in both bat and human cells.

Doxorubicin is used to kill cancer cells in the clinic. However, cancer cells that acquire high ABCB1 expression through prolonged chemotherapy exposure efflux out chemotherapeutic drugs such as doxorubicin, reducing their efficacy[7,11]. Therefore, we tested whether we could enhance doxorubicin-induced cell death in ABCB1-expressing cells by co-treating cells with doxorubicin and *Cyclo-(L-1-methyl-Trp-L-1-methyl-Trp)*. PaKiT03 parental cells were pretreated with *Cyclo-(L-1-methyl-Trp-L-1-methyl-Trp)* or verapamil for 30 minutes and then co-incubated with 1 μM doxorubicin for 48 hours. Doxorubicin treatment itself reduced the cell viability to about 60% (Fig. 5d). *Cyclo-(L-1-methyl-Trp-L-1-methyl-Trp)* or verapamil co-treatment with doxorubicin

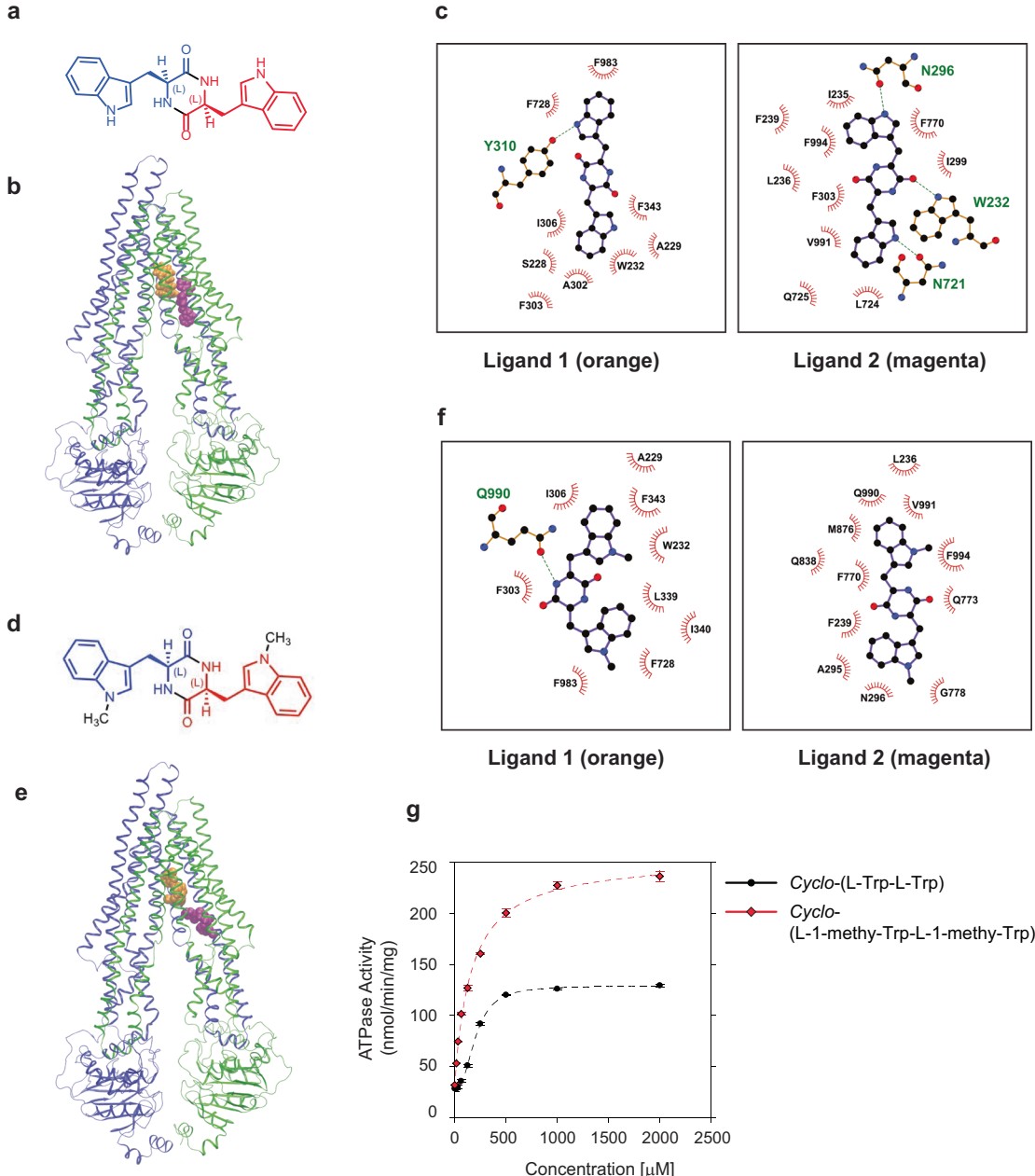

**Fig. 4 | Computational modelling of *Cyclo*-(L-Trp-L-Trp) and *Cyclo*-(L-1-methyl-Trp-L-1-methyl-Trp) binding sites on ABCB1. a** *Cyclo*-(L-Trp-L-Trp) chemical structure. **b** Docking calculation for pairs of *Cyclo*-(L-Trp-L-Trp) bound to ABCB1. Two *Cyclo*-(L-Trp-L-Trp) molecules can bind to ABCB1. Protein is shown in cartoons while each *Cyclo*-(L-Trp-L-Trp) is shown in space-fill format. The first docked compound, ligand 1, is coloured orange, and the second docked compound, ligand 2, is coloured magenta. **c** The binding site of *Cyclo*-(L-Trp-L-Trp) in a Ligplot + scheme. Corresponding residues involved in binding interactions are shown. Residues are labelled, with hydrophobic interactions indicated by red dashed arcs, and hydrogen bonds by labelled green dashed lines. **d** *Cyclo*-(L-1-methyl-Trp-L-1-methyl-Trp) chemical structure. **e** Docking calculation for pairs of *Cyclo*-(L-1-methyl-Trp-L-1-methyl-Trp) bound to ABCB1. Two *Cyclo*-(L-1-methyl-Trp-L-1-methyl-Trp) molecules can bind to ABCB1. Protein is shown in cartoons while each *Cyclo*-(L-1-methyl-Trp-L-1-methyl-Trp) is shown in space-fill format. The first docked compound, ligand 1, is coloured orange, and the second docked compound, ligand 2, is coloured magenta. **f** The binding site of *Cyclo*-(L-1-methyl-Trp-L-1-methyl-Trp) in a Ligplot+ scheme. Corresponding residues involved in binding interactions are shown. Residues are labelled, with hydrophobic interactions indicated by red dashed arcs, and hydrogen bonds by labelled green dashed lines. **g** ATPase assay of purified and reconstituted human ABCB1 protein with *Cyclo*-(L-Trp-L-Trp) and *Cyclo*-(L-1-methyl-Trp-L-1-methyl-Trp). The experiment was performed in triplicate, and the mean values ± S.D. are shown. Black line: *Cyclo*-(L-Trp-L-Trp); red line: *Cyclo*-(L-1-methyl-Trp-L-1-methyl-Trp).

further reduced the cell viability to lower than 40%, a result similar to those observed in PaKiT03 *ABCB*1 KO cells treated with doxorubicin alone (Fig. 5d). Similar results were obtained using human HEK293T-cells expressing ABCB1 (Fig. 5e) and HCT-15 cells (Fig. 5f). Together, these results indicate that *Cyclo*-(L-1-methyl-Trp-L-1-methyl-Trp) inhibits ABCB1-mediated doxorubicin efflux, and thereby sensitise doxorubicin-induced cell death in both bat and human cells.

## Regioselective benzylation of *Cyclo*-(L-Trp-L-Trp) enhances the ABCB1 inhibition

As *Cyclo*-(L-1-methyl-Trp-L-1-methyl-Trp) still requires a high concentration (0.25 mM) to inhibit human ABCB1 comparable to verapamil (5 µM) (Fig. 5c), we next explored different modifications of *Cyclo*-(L-Trp-L-Trp). Many of the known ABCB1 substrates are over 500 Daltons[25]. However, the molecular weight of *Cyclo*-(L-1-methyl-Trp-L-1-methyl-Trp)

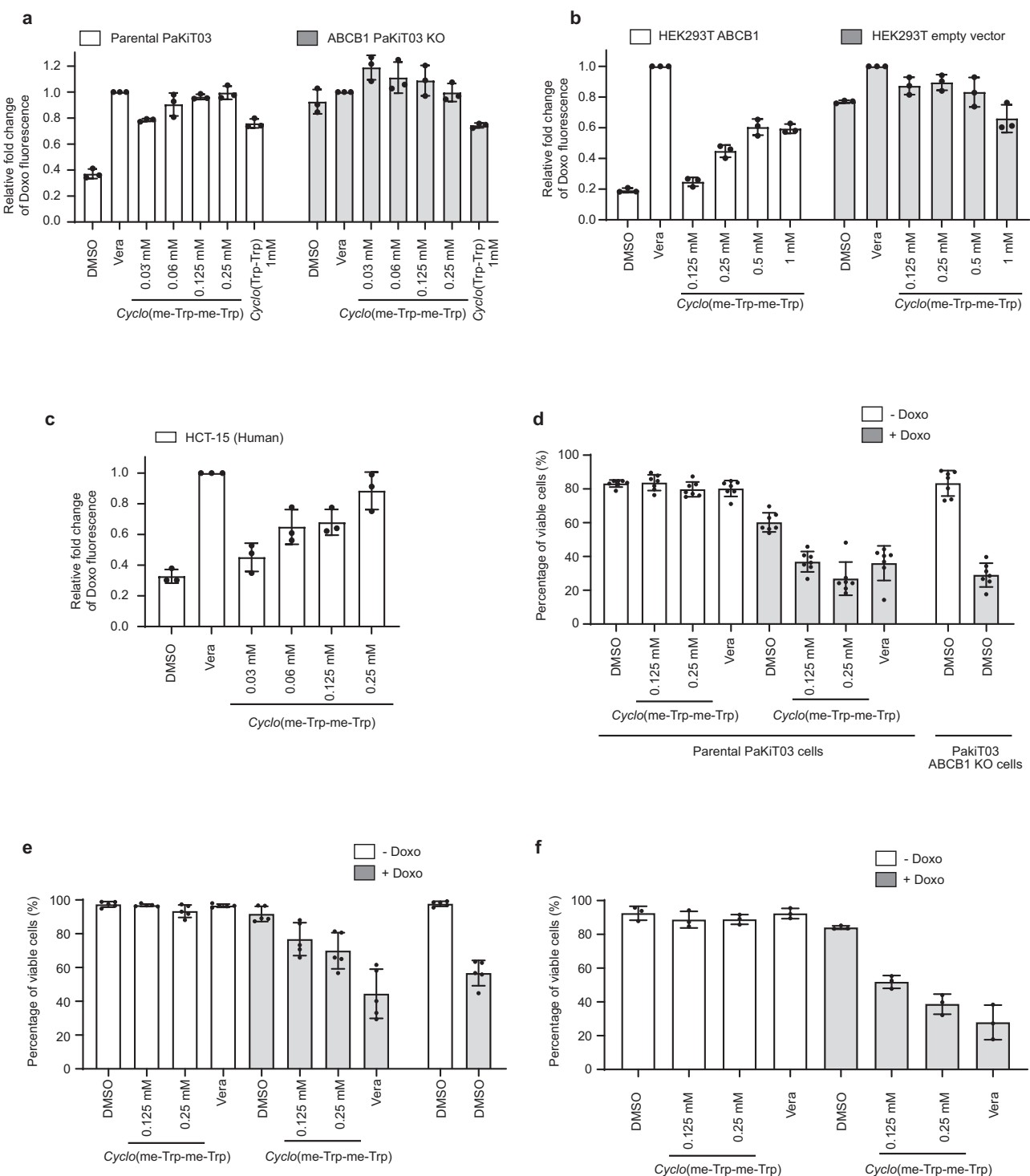

**Fig. 5 | *Cyclo*-(L-1-methyl-Trp-L-1-methyl-Trp) inhibits ABCB1 efflux and sensitises cells to doxorubicin-induced cell death. a–c** Flow cytometry analysis of doxorubicin (Doxo) accumulation. Cells were pre-treated with 5 µM verapamil (Vera), 1 mM *Cyclo*-(L-Trp-L-Trp), or the indicated dose of *Cyclo*-(L-1-methyl-Trp-L-1-methyl-Trp) before incubating with 10 µM Doxo. Bar graphs represent the mean ( ± SD of three independent experiments) fluorescent intensity of Doxo relative to the Vera-treated cells (Vera, the second bar in each cell line). **a** White bars represent parental PaKiT03 cells, while gray bars represent PakiT03 *ABCB1* KO cells. **b** White bars represent HEK293T ABCB1 expressing cells, while gray bars

represent HEK293T empty vector. **c** White bars represent HCT-15 cells. **d–f** Quantification of cell viability by Viakrome 405 staining. PaKiT03 parental and *ABCB1* KO cells **d**, HEK293T control and ABCB1 expressing cells **e**, and HCT-15 cells **f** were pre-treated with the indicated amount of *Cyclo*-(L-1-methyl-Trp-L-1-methyl-Trp) or 5 µM verapamil (Vera) for 30 minutes before adding 1 µM Doxo. The cell viability was measured after 48 hours of Doxo treatment. Bar graphs represent the mean ± SD of seven **d**, five **e**, or three **f** independent experiments. White bar: without Doxo; gray bar: with Doxo.

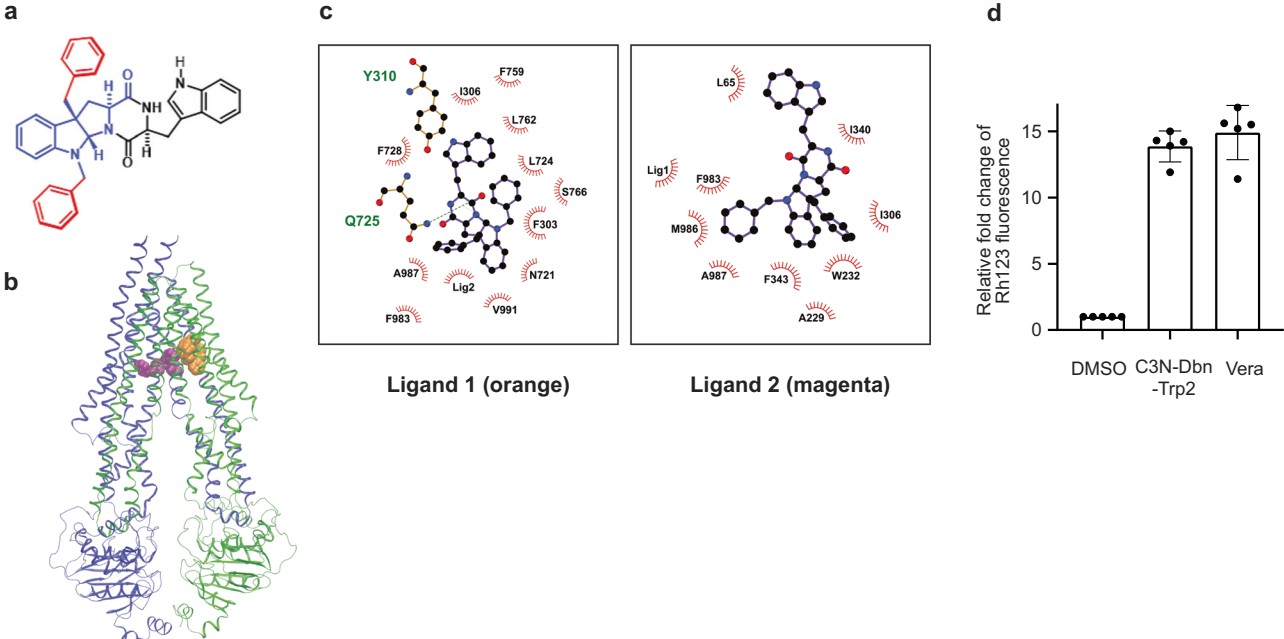

**Fig. 6 | Computational modeling of C3N-Dbn-Trp2 binding sites on ABCB1.**
**a** Chemical structure of C3N-Dbn-Trp2. **b** Docking calculation for pairs of C3N-Dbn-Trp2 bound to ABCB1. Two C3N-Dbn-Trp2 molecules can bind to ABCB1. Protein is shown in cartoons while each C3N-Dbn-Trp2 is shown in space-fill format. The first docked compound, ligand 1, is coloured orange, and the second docked compound, ligand 2, is coloured magenta. **c** The binding site of C3N-Dbn-Trp2 in a Ligplot+ scheme. Corresponding residues involved in binding interactions are shown. Residues are labelled, with hydrophobic interactions indicated by red dashed arcs, and hydrogen bonds by labelled green dashed lines. **d** Flow cytometry analysis of rhodamine 123 (Rh123) accumulation. Parental PaKiT03 cells were pretreated with 10 μM C3N-Dbn-Trp2 or verapamil (Vera) before incubating with 2.5 μM Rh123. Bar graphs represent the mean ( ± SD of five independent experiments) fluorescent intensity of Rh123 relative to the DMSO-treated cells (the first bar).

is 400.4 Daltons. In order to investigate whether any increase in size and hydrophobicity to the *Cyclo*-(L-Trp-L-Trp) scaffold could lead to better inhibition of ABCB1, we resorted to applying regioselective derivatization of *Cyclo*-(L-Trp-L-Trp) scaffold with two benzyl groups. A benzyl group is hydrophobic like a methyl group, but bulkier and also aromatic in nature. This putative molecule, named C3-N1-dibenzyl-exo-pyrroloindoline-cyclic-L-tryptophan-L-tryprophan (C3N-Dbn-Trp2) (Fig. 6a), was tested in our computational simulations. Our results indicated that two C3N-Dbn-Trp2 molecules can bind to ABCB1 (Fig. 6b). One molecule (ligand 1 in orange) occupied the access tunnel[22], and another molecule (ligand 2 in magenta) bound to the central pocket, similarly to *Cyclo*-(L-1-methyl-Trp-L-1-methyl-Trp). (Fig. 6c). Furthermore, on average across multiple simulation replicas, the C3N-Dbn-Trp2 - ABCB1 interaction exhibited more stable binding than *Cyclo*-(L-1-methyl-Trp-L-1-methyl-Trp) (Supplementary Video 1). This stability is partially due to the way the two ligands bind together via 6.8 ± 3.2 hydrophobic contacts (Table 1). Consistently, our MM-PBSA results predicted that C3N-Dbn-Trp2 binds to ABCB1 with the highest affinity of all ligands tested (Table 1 and S1). Since the results of our simulations were promising, we decided to synthesize C3N-Dbn-Trp2. Our earlier studies have established direct C3-alkylation methods to access unnatural derivatives of natural products that are tryptophan-derived diketopiperazines[29,30]. Through this method of benzylation, we purified a compound C3N-Dbn-Trp2 (Supplementary Fig. S3).

To confirm our simulations, PaKiT03 parental cells were pretreated with 10 μM of C3N-Dbn-Trp2 or verapamil for 30 minutes and then co-incubated with rhodamine 123 for 3 hours for flow cytometry analysis. 10 μM of C3N-Dbn-Trp2 inhibited the rhodamine 123 efflux similarly to 10 μM of verapamil (Fig. 6d), which is much more effective than *Cyclo*-(L-1-methyl-Trp-L-1-methyl-Trp) (125 μM) (Supplementary Fig. S4). Furthermore, PaKiT03 *ABCB1* KO cells showed accumulation of rhodamine 123, regardless of the treatment with C3N-Dbn-Trp2 (Supplementary Fig. S5), suggesting that the effect of C3N-Dbn-Trp2 in PaKiT03 parental cells is through ABCB1 inhibition. These results indicate that benzylation

of *Cyclo*-(L-Trp-L-Trp) greatly improves ABCB1 inhibition. During the synthesis of benzylated *Cyclo*-(L-Trp-L-Trp), we also obtained C3N-Dbn-Trp2 and three minor fractions of benzylated compounds (Supplementary Fig. S3). However, these minor compounds did not inhibit ABCB1 as strongly as C3N-Dbn-Trp2. (Supplementary Fig. S6). Therefore, we decided to focus on C3N-Dbn-Trp2 for further assessment.

## C3N-Dbn-Trp2 inhibits ABCB1 as effectively as verapamil

To compare the inhibitory activity of C3N-Dbn-Trp2 and verapamil more accurately, we performed titration of these compounds and assessed the accumulation of rhodamine 123 by flow cytometry in human cells. ABCB1-expressing HEK293T cells were pretreated with C3N-Dbn-Trp2 or verapamil at a series of dilutions for 30 minutes and then co-incubated with rhodamine 123 for 2 hours for flow cytometry analysis. The amount of rhodamine 123 accumulated in cells was measured, and the $IC_{50}$ values were calculated based on the extent to which rhodamine 123 efflux was inhibited by these compounds. The $IC_{50}$ of C3N-Dbn-Trp2 and verapamil for rhodamine 123 efflux were 5.9 μM and 20.5 μM respectively (Fig. 7a). Similar experiments were done using human HCT-15 cells. The $IC_{50}$ of C3N-Dbn-Trp2 and verapamil for rhodamine 123 efflux in HCT-15 cells were 2.2 μM and 7.2 μM respectively (Fig. 7b). In addition, the $IC_{50}$ of C3N-Dbn-Trp2 and verapamil for rhodamine 123 efflux in bat PaKiT03 cells were 0.9 μM and 1.0 μM respectively (Supplementary Fig. S7a). Thus, compared to verapamil, the efficacy of ABCB1 inhibition by C3N-Dbn-Trp2 was superior in human cell lines and was comparable in bat cell lines among the cell lines tested. On the other hand, we also measured the $IC_{50}$ of tariquidar, a potent and well-characterized third-generation ABCB1 inhibitor, using ABCB1-expressing human HEK293T cells, human HCT-15 cells, and bat PaKiT03 cells. The $IC_{50}$ of tariquidar for rhodamine 123 efflux in these cells were 16 nM, 58 nM, and 2.0 nM, respectively (Fig. 7a, b, and Supplementary Fig. S7a). These results indicate that C3N-Dbn-Trp2 is less potent than tariquidar, indicating the need for further structural modifications to enhance its efficacy.

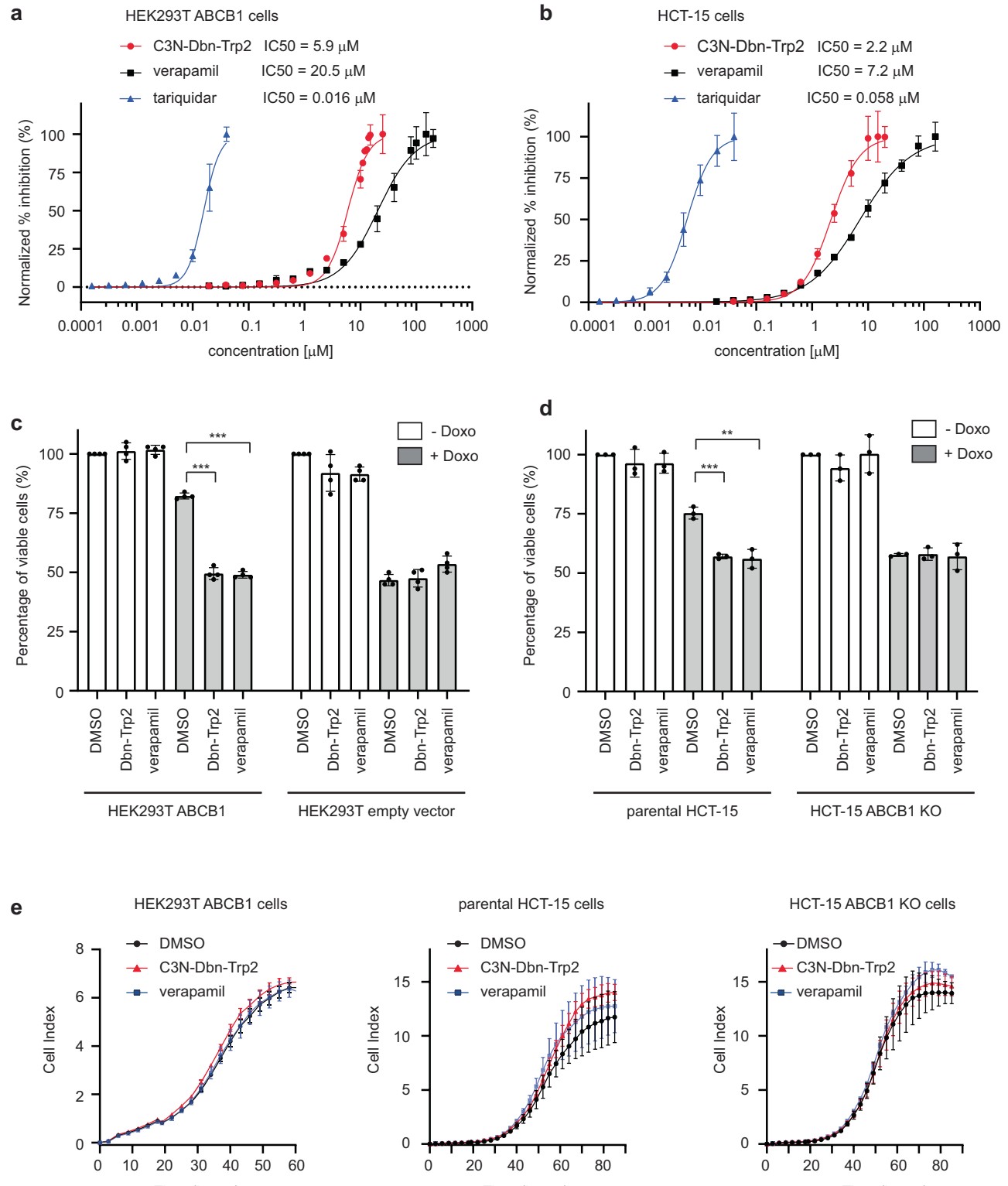

We next examined whether C3N-Dbn-Trp2 could enhance doxorubicin-induced cell death in ABCB1-expressing cells by co-treating cells with doxorubicin and C3N-Dbn-Trp2. ABCB1-expressing human HEK293T and HCT-15 cells were pretreated with C3N-Dbn-Trp2 or verapamil for 30 minutes and then co-incubated with 2.5 μM doxorubicin for 24 hours. Doxorubicin treatment itself reduced the cell viability to about 75% (Fig. 7c, d). C3N-Dbn-Trp2 or verapamil co-treatment with doxorubicin further reduced the cell viability to around 50%. This is similar to those observed in HEK293T control cells or HCT-15 *ABCB1* KO cells treated with doxorubicin alone (Fig. 7c, d, and Supplementary Fig. S8). Similar results were obtained using bat PaKiT03 parental and *ABCB1* KO cells (Supplementary Fig. S7b). C3N-Dbn-Trp2 did not affect cell viability when cells were cultured without doxorubicin (Fig. 7c, d). To further confirm this observation, we monitored the effect of C3N-Dbn-Trp2 on cell growth for two to three days without doxorubicin. There was no adverse effect of C3N-Dbn-Trp2 on cell proliferation in HEK293T or HCT-15 cells

**Fig. 7 | C3N-Dbn-Trp2 inhibits ABCB1 efflux and sensitises cells to doxorubicin-induced cell death. a, b** Flow cytometry analysis of rhodamine 123 (Rh123) accumulation. HEK293T ABCB1 expressing cells **a** and HCT-15 cells **b** were pre-treated with different concentrations of C3N-Dbn-Trp2, verapamil or tariquidar before incubating with 2.5 μM Rh123. The concentrations of the compounds that reach the maximum accumulation of Rh123 were set as 100% inhibition. Graphs represent the percentage inhibition ( ± SD of three independent experiments) compared to DMSO pre-treated cells. The half-maximal inhibitory concentration (IC$_{50}$) values were calculated using nonlinear regression. Red line (circle): C3N-Dbn-Trp2; black line (square): verapamil; blue line (triangle): tariquidar. **c, d** Quantification of cell viability by ATPlite Luminescence assay. HEK293T control and ABCB1 expressing cells **c**, and HCT-15 parental and *ABCB1* KO cells **d** were pre-treated with 5 μM of C3N-Dbn-Trp2 (Dbn-Trp2) or verapamil for 30 minutes before adding 2.5 μM doxorubicin (Doxo). The cell viability was measured after 24 hours of Doxo treatment. Bar graphs represent the mean ± SD of four **c** or three **d** independent experiments. Unpaired two-tailed Student's t-test was performed for statistical analysis (**$P < 0.01$, ***$P < 0.001$). White bar: without Doxo; gray bar: with Doxo. **e** Real-time cell proliferation analysis by xCELLigence system. HEK293T ABCB1 expressing cells, and HCT-15 parental and *ABCB1* KO cells were cultured with 5 μM of C3N-Dbn-Trp2 or verapamil for the indicated time. Graphs represent the mean cell index ± SD of four (HEK293T ABCB1 cells and HCT-15 parental cells) or five (HCT-15 *ABCB1* KO cells) independent experiments. Black line (circle): DMSO; red line (triangle): C3N-Dbn-Trp2; blue line (square): verapamil.

(Fig. 7e). Together, these results suggest that C3N-Dbn-Trp2 enhanced doxorubicin-induced cell death by inhibiting ABCB1-mediated doxorubicin efflux, but not by triggering cytotoxicity in the cells.

## C3N-Dbn-Trp2 partially inhibits ABCG2, but not ABCC1

In addition to ABCB1, other ABC family proteins, ABCC1 and ABCG2, have also been implicated in chemotherapeutic drug resistance[11]. We investigated whether C3N-Dbn-Trp2 exhibits inhibitory activity against ABCC1 and ABCG2. To achieve this, we first generated *ABCC1* CRISPR knockout cells using HEK293T and HCT-15 cells, both of which endogenously express ABCC1 (Fig. 8a, b). To specifically examine the effects of ABCC1 inhibition in HCT-15 cells, which also express ABCB1, we generated *ABCC1* knockout cells from *ABCB1* knockout HCT-15 cells (Fig. 8c). These knockout and parental cells were pretreated with C3N-Dbn-Trp2 and then co-incubated with CellTracker Green CMFDA, a fluorescent substrate of ABCC1, for flow cytometry analysis. As expected, CMFDA was accumulated by *ABCC1* knockout (Fig. 8d, e). We investigated whether C3N-Dbn-Trp2 could prevent CMFDA efflux in parental cells. Although C3N-Dbn-Trp2 clearly inhibited ABCB1 efflux activity at 10 μM (Fig. 7a, b), it failed to inhibit ABCC1 efflux activity in both HEK293T and *ABCB1*-knockout HCT-15 cells at the same concentration (Fig. 8d, e). Following a similar approach, we generated *ABCG2* knockout cells using human placental choriocarcinoma BeWo cells, which endogenously express ABCG2 (Fig. 8a, f). After pre-treating *ABCG2* knockout and parental BeWo cells with C3N-Dbn-Trp2, we co-incubated them with the fluorescent ABCG2 substrate, Hoechst 33342, for flow cytometry analysis. While the well-known ABCG2 inhibitor Ko143 induced Hoechst 33342 accumulation comparable to levels observed in *ABCG2* knockout cells, C3N-Dbn-Trp2 only partially inhibited ABCG2 efflux activity at 10 μM (Fig. 8g). Taken together, C3N-Dbn-Trp2 exhibits the most potent efflux inhibition against ABCB1 compared to ABCC1 and ABCG2. This selectivity is supported by computational simulations demonstrating a stable interaction between C3N-Dbn-Trp2 and ABCB1 (Fig. 6 and Table 1 and S1).

## Discussion

Following the discovery of ABCB1 as a multi-drug resistant protein in 1976[31] and its subsequent cloning in 1985[32] and 1986[33,34] as the first members of the ABC transporter family of 49 proteins[35], ABCB1 has been extensively studied for its role in anticancer drug resistance and pharmacokinetic properties of various drugs due to its drug efflux capabilities. ABCB1 overexpression is not limited to specific cancer types, and therefore ABCB1 inhibitors could potentially treat a broad range of cancers[11]. However, none of these inhibitors have been successful in clinical trials due to their insufficient specificity in treating cancer patients[11]. One reason for this lack of specificity is the limited knowledge of the substrate structure that ABCB1 prefers to bind. In this study, we propose a model suggesting that the tryptophan structure could serve as a lead molecule for the development of ABCB1 inhibitors, based on biological analyses combined with computer-generated molecular modelling of binding between the ligand and ABCB1.

ABCB1 is known to efflux a wide range of exogenous substrates, including chemotherapeutic agents, antibiotics, steroids, lipids, and other xenobiotic compounds[36,37]. Substrate promiscuity and polyspecificity are known features of ABCB1, which has hundreds of known substrates whose numbers continue to grow. This makes it challenging to design efficient competitive inhibitors[36,37]. Although structural information of ABCB1 has been obtained by many laboratories due to its importance as a multi-drug resistant protein[36,37], the promiscuous substrate-binding sites of ABCB1 make it difficult to predict substrate structure. The molecular basis of this unusual property of ABCB1 has been a matter of debate for a long time. However, in the past five years, structural information of ABCB1 with its substrates has provided breakthroughs in this debate using cryoEM and double electron-electron resonance (DEER), leading to a better understanding of ABCB1 ligand binding[18–22]. We used the published ABCB1 structures to modify tryptophan step by step, demonstrating the proof of concept that tryptophan and its cyclic dimer are one of the promising lead structures that ABCB1 prefers to bind. Our first modification was a diketopiperazine formation to generate a cyclic dimer of tryptophan *Cyclo*-(L-Trp-L-Trp) that is naturally produced by bacteria[26]. *Cyclo*-(L-Trp-L-Trp) was further modified by adding alkyl groups stereoselectively and regioselectively to enhance lipophilicity because our computational modelling showed that *Cyclo*-(L-Trp-L-Trp) binds in a lipophilic area of ABCB1. Increasing $sp^3$ centres on aromatic rings such as an indole, by installing branched substituents, to generate a pyroloindoline scaffold from a tryptophan ring, offers numerous advantages. This change usually enhances the bioactivity of the small molecule, as evident from numerous biologically active natural products that possess this puckered ring system, such as ardeemin[26,27]. Similar to natural product scaffolds, we anticipated that the puckered ring system with additional lipophilic groups such as an aromatic ring could enhance favourable interactions at the binding site of the ABCB1 protein. Increasing lipophilicity can be also achieved by simple alkylation of either (or both) indole nitrogen[38] or amino nitrogen atoms of tryptophan prior to *Cyclo*-(L-Trp-L-Trp) formation. Alkyl groups can be simple primary groups such as methyl, ethyl, propyl, butyl, and isobutyl, as well as benzyl to probe for π-interactions. Both symmetrical analogues and unsymmetrical analogues can be generated. Using this strategy, we successfully generated C3N-Dbn-Trp2 and enhanced its activity for ABCB1 inhibition. As our studies originated from the observation of the accumulation of natural form, L-tryptophan, we have accordingly restricted our studies to compounds with the same sense of chirality. In addition, the synthetic methods we employed for the preparation of C3N-Dbn-Trp2 from Cyclo-(L-Trp-L-Trp), did not result in any racemization, yielding an enantiomerically pure compound. The results from C3N-Dbn-Trp2 indicate that rigidification of the molecule by annulation between indole and *Cyclo*-(L-Trp-L-Trp) has a large beneficial effect on its activity. Interestingly, some natural products, such as ardeemin[39], bear a strong structural resemblance to *Cyclo*-(L-Trp-L-Trp)[40] but with a more rigid structure. Therefore, the structure of natural products such as ardeemin could be used as a lead to further inspire the design of inhibitors.

Our initial idea of using the tryptophan structure for developing ABCB1 inhibitors originated from the metabolic analysis of bat cells. Bats are the second-largest order of mammals and the only mammals capable of powered-wing flight[1,41]. Despite a fast metabolic rate due to flight, all bats are long-lived and generally have very low cancer rates[1,2]. However, the mechanism underlying their cancer resistance is unclear. The elevated

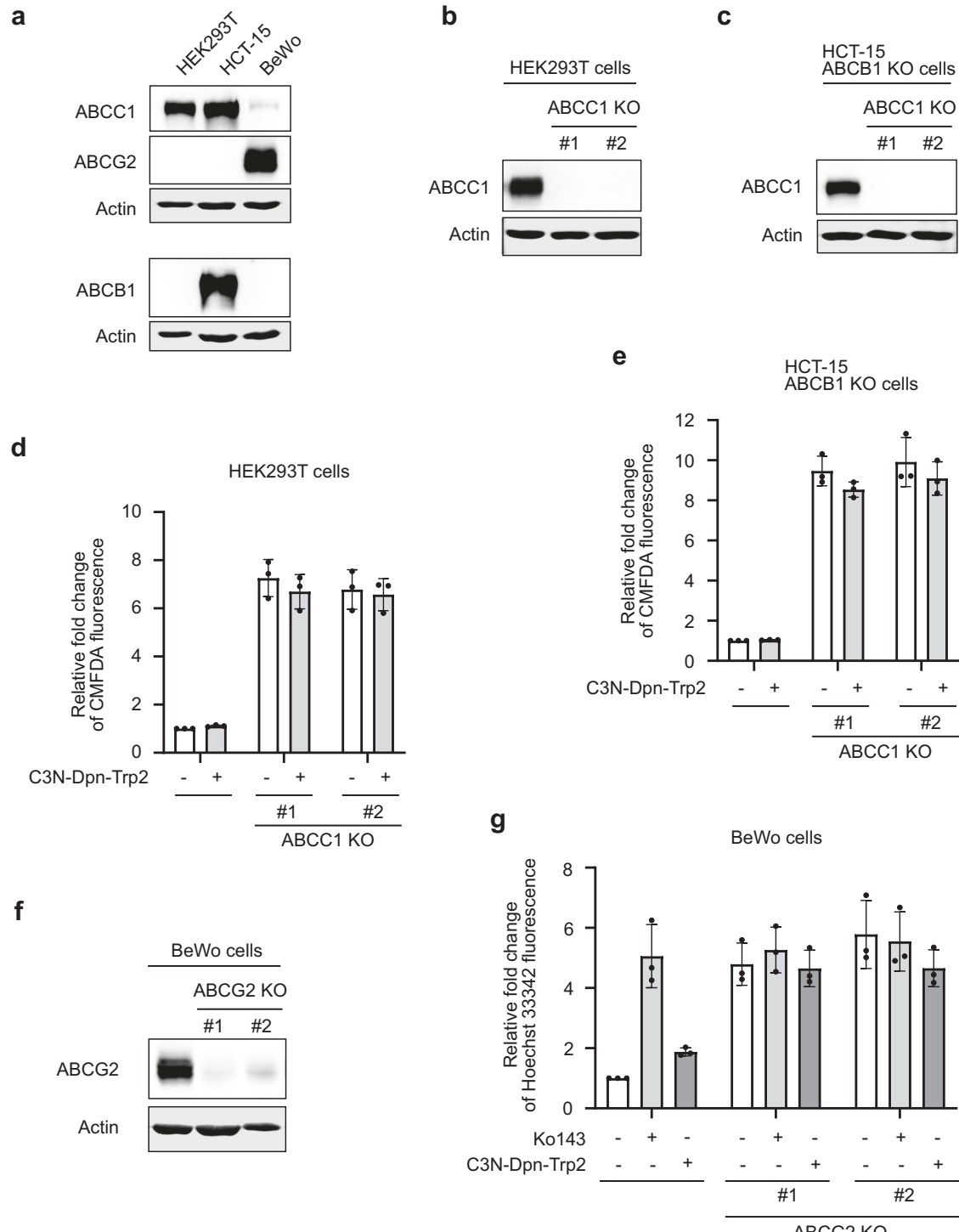

**Fig. 8 | C3N-Dbn-Trp2 partially inhibits ABCG2, but not ABCC1. a–c** Western blot analysis of ABC family proteins in the indicated parental cells and *ABCC1* knockout cells. Actin was used as a loading control. **d, e** Flow cytometry analysis of CMFDA accumulation. HEK293T parental and *ABCC1* knockout cells **d**, and HCT-15 *ABCB1* knockout cells and *ABCB1/ABCC1* double knockout cells **e** were pre-treated with 10 μM C3N-Dbn-Trp2 before incubating with 1 μM CMFDA. Bar graphs represent the mean ( ± SD of three independent experiments) fluorescent intensity of CMFDA relative to the DMSO-treated parental cells (the first bar). **f** Western blot analysis of ABCG2 in BeWo parental cells and *ABCG2* knockout cells. Actin was used as a loading control. **g** Flow cytometry analysis of Hoechst 33342 accumulation. BeWo parental and *ABCG2* knockout cells were pre-treated with 10 μM Ko 143 or C3N-Dbn-Trp2 before incubating with 3 μM Hoechst 33342. Bar graphs represent the mean ( ± SD of three independent experiments) fluorescent intensity of Hoechst 33342 relative to the DMSO-treated parental cells (the first bar).

expression of ABCB1 in bats may protect them not only from genotoxic substances in their environment[42–44], but also from harmful metabolites produced in bat cells. Most current ABCB1 inhibitors are ABCB1 substrates that act as competitive inhibitors[11,45]. Although endogenous metabolites that

ABCB1 can efflux are not well-studied, cortisol and aldosterone are known examples[46], and both hormones are produced and secreted by the adrenal gland, which expresses high levels of ABCB1 in humans. The clinical trial failures with synthetic compounds have prompted research into naturally

occurring ABCB1 inhibitors, which might be safer and less toxic. ABCB1 activity can be inhibited by garlic[47], green tea[48], ginseng[49], and some fruits[50], but the inhibitory substances within these foods, and their potential therapeutic application are unknown. While our results did not confirm tryptophan as an ABCB1 substrate for bat ABCB1, our data showed that the tryptophan structure can serve as a lead molecule for designing ABCB1 inhibitors. Building on this, endogenous metabolites within tryptophan degradation pathways or other derivatives, such as serotonin, melatonin, and tryptamine may also be tested for ABCB1 inhibition in the future.

C3N-Dbn-Trp2 represents the first member of a unique class of efflux pump inhibitors, inspired by bat studies and based on a tryptophan structure. As such, we believe this conceptual advance will be of interest to the medicinal chemistry and pharmacology fields as the initial report. Nevertheless, being the first of its kind, the current studies on C3N-Dbn-Trp2 are not without limitations. Firstly, although the IC$_{50}$ value of C3N-Dbn-Trp2 for the efflux of rhodamine 123 is slightly better than that of the first-generation inhibitor verapamil, it is still significantly lower than the IC$_{50}$ value of the third-generation ABCB1 inhibitor, tariquidar. Thus, further modifications of C3N-Dbn-Trp2 should be extensively explored to improve the IC$_{50}$ values. Secondly, although C3N-Dbn-Trp2 is derived from the natural compound Cyclo-(L-Trp-L-Trp), it does not necessarily imply that Cyclo-(L-Trp-L-Trp) and its derivatives are safer in humans. Even though we have not observed the adverse effects of C3N-Dbn-Trp2 on cell proliferation in a cell culture system, this does not guarantee its safety when administered in vivo. Therefore, it will be important to validate its safety in comparison to other ABCB1 inhibitors using an in vivo model. Thirdly, we need to ensure that C3N-Dbn-Trp2 and its derivatives can efficiently promote the killing of cancer cells with chemotherapeutic drugs while minimizing side effects on normal cells expressing ABCB1 in the body. Although ABCB1 is expressed in a limited number of human normal tissues and at much lower levels than in certain drug-resistant cancers, it is necessary to confirm that ABCB1 inhibition does not affect normal cell functions. Lastly, the failure of clinical trials for ABCB1 inhibitors can be attributed to several factors, including cellular detoxification pathways[11,51]. Therefore, assessing the stability of C3N-Dbn-Trp2 in a mouse model is crucial, as both instability and over-stability are problematic in vivo.

Developing new ABCB1 inhibitors has strong long-term clinical implications. First, clinical trials testing synthetic ABCB1 inhibitors in drug-resistant cancers have been unsuccessful for 30 years[11]. A novel strategy to develop ABCB1 inhibitors is urgently needed to overcome drug resistance in cancers[52–54]. Second, ABCB1 plays a critical role in drug efflux at the blood-brain barrier[55,56] that can interfere with the delivery of chemotherapeutic drugs to treat brain cancers[11,57]. Therefore, administering chemotherapy alongside an ABCB1 inhibitor is an attractive therapeutic approach, particularly for tissues or cancers with high ABCB1 activity. Third, besides cancer therapy, ABCB1 inhibitors could be used for neurodegenerative and psychiatric diseases when the therapeutic drugs are ABCB1 substrates, such as pergolide, bromocriptine, pramipexole, olanzapine, risperidone, and 9-OH risperidone[58–60]. These drugs have had limited success, and drug delivery to the brain remains a major challenge. Transient administration of these drugs together with a specific ABCB1 inhibitor could potentially enable efficient delivery of these drugs to the brain, reduce the required dosage, and minimize off-target side effects. Fourth, HIV-1 protease inhibitors such as ritonavir, saquinavir, and indinavir, are ABCB1 substrates[61–63]. Similar to cancer cells that elevate ABCB1 expression during chemotherapy, T-cells also elevate ABCB1 expression during prolonged treatment with HIV-1 inhibitors and gain drug resistance[64–66]. Thus, specific ABCB1 inhibitors could be clinically valuable for HIV patients[64–66]. Fifth, the multi-drug resistance of bacteria is an urgent problem in the clinic. Drug-resistant bacteria express high levels of ABCB1 orthologs[67]. ABCB1 inhibitors could be tested for their ability to improve the efficacy of antibiotics.

In conclusion, by utilizing bat cells that naturally express high levels of ABCB1, we demonstrated the proof of concept that the tryptophan structure is a promising starting point for developing ABCB1 inhibitors. Assisted by computational modelling to analyze interactions between ABCB1 and substrates, along with recently published ligand-ABCB1 binding data, we developed a compound C3N-Dbn-Trp2 derived from tryptophan. This compound achieved similar specificity and efficacy to the widely known ABCB1 inhibitor, verapamil in restoring sensitivity to chemotherapeutic drugs in drug-resistant cancer cells expressing ABCB1. Our unique approach to the development of ABCB1 inhibitors may help to achieve the long-standing goal of using ABCB1 inhibitors against drug-resistant cancers and other diseases.

## Methods

### Cell culture

PaKiT03 cells were derived from *P. alecto* kidney and immortalized with large T-antigen[68]. HCT-15, HEK293T, and BeWo cells were purchased from American Type Culture Collection (ATCC). PakiT03, HCT-15, and HEK293T cells were grown in Dulbecco's modified Eagle's medium (DMEM) with 10% fetal bovine serum and penicillin/streptomycin, and BeWo cells were grown in Ham's F-12K (Kaighn's) medium with 10% fetal bovine serum and penicillin/streptomycin. All the cells were cultured in a 37°C incubator with 5% CO$_2$.

### Reagents

Doxorubicin hydrochloride (cat# D4035), verapamil hydrochloride (cat# V4629), and L-tryptophan (cat# T0254) were obtained from Sigma-Aldrich. Rhodamine 123 (cat# R302) and CellTracker™ Green CMFDA Dye (cat# C2925) were purchased from Thermo Fisher Scientific. Hoechst 33342 (ab228551) was purchased from Abcam. *Cyclo*-(L-Trp-L-Trp) (cat# 23494), *Cyclo*-(L-Leu-L-Trp) (cat# 24942), *Cyclo*-(L-Leu-L-Pro) (cat# 24941), *Cyclo*-(L-Trp-L-Pro)/Brevianamide F (cat# 23493) were purchased from Cayman Chemical. Tariquidar (cat# HY-10550) and Ko 143 (cat# HY-10010) were purchased from Med Chem Express. *Cyclo*-(5-OH-Trp-5-OH-Trp) (MW 404.42) and *Cyclo*-(L-1-methyl-Trp-L-1-methyl-Trp) (MW 400.47) were synthesized by KE biochem co., LTD.

### C3N-Dbn-Trp2 synthesis

C3N-Dbn-Trp2 was synthesized by benzylation of *Cyclo*-(L-Trp-L-Trp). Details of synthesis, purification, and analysis are provided in the Supplementary information.

### Western blotting and antibodies

Cells were lysed with sodium dodecyl sulfate (SDS) lysis buffer (50 mM Tris–HCl pH 6.8, 2% SDS, 10% glycerol). Lysates were then quantified and analyzed by Western blotting. The antibodies used are as follows: H2AX (A303-837A, Bethyl Laboratories, Inc.), γH2AX (05-636, Millipore), tubulin (ab44928, Abcam), actin (MAB1501, Millipore), ABCB1 (GTX23364, GeneTex; sc-55510, Santa Cruz), ABCC1 (67228-1-Ig, Proteintech), ABCG2 (sc-58222, Santa Cruz), horseradish peroxidase (HRP)-conjugated secondary antibody (Jackson Immuno Research laboratories), and fluorescence-labeled secondary antibodies (Thermo Scientific Scientific). Fluorescence Detection was performed using LI-COR Odyssey (LI-COR Biosciences). HRP detection was done with HRP substrates (Thermo Scientific Scientific). Uncropped blot images are provided as Supplementary Fig. S9-13.

### Fluorescent dye accumulation assays

Cells were pre-treated with DMSO (a negative control) or the indicated compounds for 30 minutes and then co-incubated with the indicated fluorescent dye/compound for 2–3 hours. The fluorescent dyes were used at 2.5 μM (rhodamine 123), 1 μM (CMFDA), 3 μM (Hoechst 33342), or 10 μM (doxorubicin). Only when testing ABCC1 efflux activity, cells were pre-treated with the indicated compounds for 2 hours and then co-incubated with 1 μM (CMFDA) for 30 minutes. The stained cells were trypsinized, washed once with 1x PBS, and suspended in 1× PBS for flow cytometry. The fluorescence intensity was measured by MACSQuant Analyser 10 (Miltenyi Biotec). Data were analyzed by FlowJo software.

### Viakrome 405 cell viability assay

Cells were seeded at 15,000 cells per well in 6-well plates. One day after seeding, cells were pre-treated with either DMSO or the indicated compounds for 30 minutes, and then incubated without or with 1 μM doxorubicin for an additional 48 hours. All the floating and attached cells were collected, washed twice with 1x PBS, and stained with Viakrome 405 as instructed (Beckman Coulter). The fluorescence intensity of Viakrome 405 was measured by MACSQuant Analyser 10 (Miltenyi Biotec). Dead cells are strongly stained with Viakrome 405. Data were analyzed by FlowJo software.

### ATPlite Luminescence Assay (cell viability assay)

Cells were seeded at 12,500 cells per well in 6-well plates. One day after seeding, cells were pre-treated with either DMSO or the indicated compounds for 30 minutes, and then incubated without or with 2.5 μM doxorubicin for an additional 24 hours. The amount of ATP was measured by Perkin Elmer ATPlite Luminescence assay as instructed. The ATP amounts of non-drug treated cells were set to 100% cell viability and the relative cell viability is shown.

### Metabolomics screening and analysis

Metabolites of PaKiT03 cells were measured by HMT capillary electrophoresis-mass spectrometry (CE-MS) analysis (Human Metabolome Technologies Inc., Yamagata, Japan). For metabolite extraction, $2-3 \times 10^6$ cells of parental PaKiT03 and each *ABCB1* KO clone were used according to the HMT's instruction. Briefly, cells were cultured in 10 cm dishes and washed twice with 5% mannitol solution before being treated with methanol-containing internal standard solution. The metabolite extracts were then centrifuged at $2300 \times g$ for 5 min at 4 °C, and supernatants were further centrifugally filtered at $9100 \times g$ for 120 minutes at 4 °C. Lastly, filtrates were dried by SpeedVac Concentrator SPD1010 (Thermo Fisher Scientific) and submitted to HMT for analysis. The amounts of L-tryptophan per $10^6$ cells were calculated based on the results from the HMT analysis. The results of CE-MS were further analyzed by Ingenuity Pathway Analysis (QIAGEN).

### CRISPR knockout of *ABCB1* in PaKiT03 cell line

Exon sequence close to the 5′-end of the *ABCB1* gene was submitted to the online software (http://tools.genome-engineering.org) to obtain the candidate gRNA target sequences. The top hits were further subjected to blast with the *P.alecto* genome to exclude the candidate gRNAs with a high probability of off-target effects. The one gRNA sequence (5′-TACA-GATCTCTCGGACAACC-3′) was chosen based on the scores for higher editing efficiency and lower off-target effects. These sequences were cloned into pSpCas9 (BB) -2A-GFP plasmid (Addgene plasmid ID: 48138) and used for transfection for the knockout as described[69]. The single-cell clones were isolated and knockout of *ABCB1* was validated using sequencing and Western Blotting analysis.

### CRISPR knockout of *ABCB1* in HCT-15 cells

The candidate gRNA sequences for human *ABCB1* were searched by Benchling (https://www.benchling.com/). The one target sequence (5′-CTTGAAGGGGACCGCAATGG-3′) was chosen based on the scores for higher editing efficiency and lower off-target effects. sgRNA of *ABCB1* was synthesized by Integrated DNA Technologies, Inc. and electroporated together with Cas9 protein (Thermo Fisher Scientific) in HCT-15 cells using Neon Electroporation System (Thermo Fisher Scientific). Cells were then stained with 2.5 μM rhodamine 123 for sorting by flow cytometry. Two distinct populations were observed, and only high Rh123-accumulating cells were collected as the *ABCB1* knockout pool of cell population and used as HCT-15 *ABCB1* knockout cells. This cell population was further assessed for *ABCB1* knockout by sequencing and Western Blotting analysis. *ABCB1* knockout single-cell clones were also isolated and validated using sequencing and Western blotting analysis. One of the single-cell clones was used for further editing to knock out *ABCC1* using CRISPR.

### *ABCC1* CRISPR knockout cells

The gRNA sequences for human *ABCC1* were selected from Predesigned Alt-R™ CRISPR-Cas9 guide RNA (Integrated DNA Technologies, Inc.). Target sequences for *ABCC1* #1 are 5′-TATCTCTCCCGACATGACCG-3′ and for *ABCC1* #2 are 5′-TTCAGAACACGGTCCTCGTG-3′. The sgRNAs were electroporated together with Cas9 protein (Thermo Fisher Scientific) in HEK293T cells or HCT-15 *ABCB1* knockout single-cell clone using Neon Electroporation System (Thermo Fisher Scientific). *ABCC1* knockout pool of cells were then stained with 1 μM CMFDA for sorting by flow cytometry. Two distinct populations were observed, and only high CMFDA-accumulating cells were collected as the *ABCC1* knockout cell populations. These cell populations were further assessed for *ABCC1* knockout by Western Blotting analysis.

### *ABCG2* CRISPR knockout cells

The gRNA sequences for human *ABCG2* were selected from Predesigned Alt-R™ CRISPR-Cas9 guide RNA (Integrated DNA Technologies, Inc.) or searched by Benchling (https://www.benchling.com/). Target sequences for *ABCG2* #1 are 5′- GGTCATTGGAAGCTGTCGCG -3′ and for *ABCG2* #2 are 5′- ACCTGGTCTCAACGCCATCC-3′. The sgRNAs were electroporated together with Cas9 protein (Thermo Fisher Scientific) in BeWo cells using Neon Electroporation System (Thermo Fisher Scientific). *ABCG2* knockout pool of cells were then stained with 3 μM Hoechst 33342 for sorting by flow cytometry. Two distinct populations were observed, and only high Hoechst 33342-accumulating cells were collected as the *ABCG2* knockout cell populations. These cell populations were further assessed for *ABCG2* knockout by Western Blotting analysis. During cell expansion, we noticed that the *ABCG2* knockout population appears to grow slower than the unedited population. Therefore, by the time the cells were expanded enough for the assay, the cells collected as *ABCG2* knockout cell populations showed traces of ABCG2 protein expression. To exclude the effect of ABCG2 expression in these knockout cells, we gated to high Hoechst 33342-accumulating cell populations in *ABCG2* knockout BeWo cells when analyzing the ABCG2 efflux activity by flow cytometry.

### Generation of HEK293T ABCB1 cells

HEK293T cells were transfected with either the empty vector or human ABCB1 expressing plasmid (pcDNA3.1-ABCB1-TEV-His-FLAG) previously described[70] using ScreenFect A (Wako). Two days after transfection, G418 (InvivoGen) at 2 mg/ml was added to the cells for selection. The selection process was terminated after control cells without plasmid transfection were killed by G418 treatment. Surviving cells with the empty vector transfection were then expanded and used as HEK293T empty vector cells. Surviving cells with ABCB1 plasmid transfection were expanded and stained with 2.5 μM rhodamine 123 for sorting by flow cytometry. Two distinct populations were observed, and only a low Rh123-accumulating cell population was collected and used as HEK293T ABCB1 cells. This cell population was further assessed for human ABCB1 expression by Western Blotting.

### ATPase assay for ABCB1 activity

Human ABCB1 was expressed and purified according to the previous report with minor modifications[70,71]. Briefly, FreeStyle 293 F cells (Thermo Fisher) were transfected with the plasmid expressing human ABCB1-FLAG by PEI-Max mediated method (Polysciences). Cells expressing human ABCB1-FLAG were solubilized with 0.8% (w/v) of n-dodecyl-beta-maltoside (DDM) (Anatrace) and purified with anti-FLAG M2 agarose (Sigma-Aldrich). Purified ABCB1 was reconstituted into egg lecithin liposome (FUJI FILM Wako Chemicals). ATPase reaction was carried at 37 °C for 30 minutes in the presence of 3 mM ATP and 5 mM $MgCl_2$ and released ADP was measured by HPLC[72]. Data were fitted to the Michaelis–Menten equation or Hill equation by using KaleidaGraph software.

### xCELLigence cell proliferation assay

Cells were seeded at 8000 cells (HEK293T ABCB1 cells) or 15,000 cells (HCT-15 cells) per well in a 96-well electronic microplate (Agilent). One day

after seeding, the indicated compounds were added to the wells, and the cellular impedance was measured every 3 hours for 60–85 hours using xCELLigence RTCA SP (Agilent) and expressed as Cell Index for cell proliferation.

## Modelling and simulations

We docked the ligands *Cyclo*-(L-Trp-L-Trp), *Cyclo*-(L-1-methyl-Trp-L-1-methyl-Trp) and C3N-Dbn-Trp2, along with control compound *Cyclo*-(L-Leu-L-Pro), to the human ABCB1 transporter (pdb-ID: 6QEX)[21] using Autodock Vina[73], after removing the antibody bound to the transporter. We initiated simulations from two different setups, as a long dynamic loop connecting the N- and C-terminal domains spanning residues 631 to 693 was not resolved in the experimental structure, and we decided to test its effect upon ligand binding. In the first setup, we reconstructed this loop, modelling it using the program GalaxyFill[74] as described in a previous study[75]. For comparison in an alternative setup, the loop was not reconstructed, and instead, the termini of residues 630 and 694 were terminated using standard patching groups (truncated model). Two ligand molecules were docked subsequently for each compound. All titratable residues were set to their standard states at neutral pH. The protein/drug complex (PDB-ID: 6QEX, including docked ligand) was subsequently inserted into a realistic mammalian lipid bilayer according to a previous study[76] and solvated with TIP3P water using the CHARMM-GUI[77,78]. The orientation of the protein in the bilayer was predicted using the OPM database[79]. NaCl ions were added to a concentration of 0.15 M, with additional counter ions added to ensure system neutrality. Molecular dynamics (MD) simulations were performed using GROMACS (version 2018)[80], with the CHARMM36m force field for proteins[81] and lipids[82]. Parameters for the ligands were obtained via the CHARMM General Force Field[83] using the ligand reader module[84] of the CHARMM GUI[78]. After equilibration for 10 nanoseconds (ns) applying position restraints on protein backbone atoms and all ligand atoms, unrestrained simulations were carried out for 100 ns at 310 K and 1 bar, keeping pressure and temperature constant using the Parrinello-Rahman barostat[85] and the Nosé-Hoover[86], respectively. All simulations were carried out in triplicate. Van der Waals interactions were calculated using a cut-off of 12 Å, switched after 10 Å. For the calculation of electrostatic interactions, the particle mesh Ewald approach (PME) was used with a real space cut-off of 12 Å. We assessed the binding free energy of the ligands based on the MM-PBSA approach using the program *g_mmpbsa*[87]. We calculated the binding free energy for both ligands simultaneously using an internal and external dielectric constant of 2 (internal) and 80 (external), respectively. Molecular graphics were visualized and plotted using VMD[88]. LigPlot+[89] was used to visualize the binding sites of the chemical compounds bound to the protein.

## Statistical analysis

All statistical analyses in this study were conducted using the unpaired student's two-sided *t*-test.

## Reporting summary

Further information on research design is available in the Nature Portfolio Reporting Summary linked to this article.

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

## Acknowledgements

This work was supported by Duke-NUS Signature Programme Block Grant, the Singapore Ministry of Health's National Medical Research Council grants (NMRC/OFIRG/15nov049/2016, COVID19TUG21-0098 / MOH-000798, NMRC/OFIRG/MOH-000639), and Singapore Ministry of Education Academic Research Fund Tier 2 Grant (MOE-T2EP30120-0012) to K.I., by DST-SERB Award # CRG-2020-005008, NSF RUI award #1709655, and IISER Tirupati to R.V., by JSPS KAKENHI (# JP18K19176) to Y.K., by Singapore National Research Foundation (NRF2012NRF-CRP001-056) to L-F.W., and by the support of BII (A*STAR) core funds to P.J.B. and A.K. Computational resources were provided by the National Supercomputing Centre (NSCC) Singapore.

## Author contributions

J.Y.P.K., Y.I., M.O., S.I., S.R.K., W.Y., P.W.C.Y., K.C, P.S.K., and Y.K. performed experiments. A.K. and P.J.B. performed computational modelling. H.M., D.M.V., T.M.K., and R.V. synthesized C3N-Dbn-Trp2. J.Y.P.K., Y.I., A.K., R.W.B., Y.K., R.V., P.J.B., and K.I. designed the experiments and analyzed the results. L.-F.W. provided critical advice and consultation. J.Y.P.K., Y.I., A.K., R.W.B., Y.K., R.V., P.J.B., and K.I. wrote the manuscript.

## Competing interests

The authors declare no competing interests.
