## [Peer Review File · Communications Chemistry]

This manuscript has been previously reviewed at another Nature Portfolio journal. This document only contains reviewer comments and rebuttal letters for versions considered at *Communications Chemistry*.REVIEWERS' COMMENTS:

Reviewer #1 (Remarks to the Author):

The authors have effectively addressed the comments raised previously. The revised version is substantially improved.

Review of comments raised by reviewer 3:

Comment 1: The authors satisfactorily addressed the comments by adding relevant changes to the revised manuscript.

The authors should cite a ref for (Lastly, the failure of clinical trials for ABCB1 inhibitors can be attributed to several factors, 487 including alternative cellular detoxification pathways) in Lines 486-489.

Comment 2: This was addressed by providing a snapshot structure-based design approach using cryo-EM structures of relevant transporters. Also, given information as to utility of this approach to design new ABCB1 inhibitors.

No further comments on this.

Comment 3: The authors have now included more physiological environment while simulating ABCB1 and their cyclo inhibitor. Thus, the quality of computational predictions has enhanced compared to the previously used less physiological environment. I also agree that computational predictions should always be concluded with caution, which the authors have attempted. Also, I like the cautionary note now added in lines 232-236.

Docking comment: The authors have answered this comment by performing simulations by using suggested PDB structures and concluded that their original choice gave the best results that aligns with their experimental observations.

Comment 4: in vivo study not needed for this exploratory study wherein lead compound still need refinement before it can be suitable for animal studies. The authors have now included known P-gp inhibitor, tariquidar, as a positive control.

Comment 5: The authors have optical rotation (Suppl. compound details) as proof to demonstrate enantiomeric purity.

Reviewer #1 (Remarks to the Author):

The authors have effectively addressed the comments raised previously. The revised version is substantially improved.

We appreciate the reviewer for taking the time to thoroughly review our manuscript.

Review of comments raised by reviewer 3:

Comment 1: The authors satisfactorily addressed the comments by adding relevant changes to the revised manuscript.

The authors should cite a ref for (Lastly, the failure of clinical trials for ABCB1 inhibitors can be attributed to several factors, 487 including alternative cellular detoxification pathways) in Lines 486-489.

We thank the reviewer for the suggestion to include the reference. Accordingly, we included the reference.

Comment 2: This was addressed by providing a snapshot structure-based design approach using cryo-EM structures of relevant transporters. Also, given information as to utility of this approach to design new ABCB1 inhibitors.

No further comments on this.

We appreciate the reviewer's positive assessment of this point.

Comment 3: The authors have now included more physiological environment while simulating ABCB1 and their cyclo inhibitor. Thus, the quality of computational predictions has enhanced compared to the previously used less physiological environment. I also agree that computational predictions should always be concluded with caution, which the authors have attempted. Also, I like the cautionary note now added in lines 232-236.

Docking comment: The authors have answered this comment by performing simulations by using suggested PDB structures and concluded that their original choice gave the best results that aligns with their experimental observations.

We are grateful for the reviewer's acknowledgment of the improvement and strengths of this section.

Comment 4: in vivo study not needed for this exploratory study wherein lead compound still need refinement before it can be suitable for animal studies. The authors have now included known P-gp inhibitor, tariquidar, as a positive control.

We thank the reviewer for their understanding regarding the exclusion of in vivo studies at this early stage of lead compound development.

Comment 5: The authors have optical rotation (Suppl. compound details) as proof to demonstrate enantiomeric purity.

We thank the reviewer for the favorable comments regarding this aspect of the manuscript.